# Value of Non-Coding RNA Expression in Biofluids to Identify Patients at Low Risk of Pathologies Associated with Pregnancy

**DOI:** 10.3390/diagnostics14070729

**Published:** 2024-03-29

**Authors:** Anne-Gael Cordier, Elie Zerbib, Amélia Favier, Yohann Dabi, Emile Daraï

**Affiliations:** Department of Obstetrics and Reproductive Medicine, Sorbonne University, Hôpital Tenon, 4 Rue de la Chine, 75020 Paris, France; anne-gael.cordier@aphp.fr (A.-G.C.); yohann.dabi@aphp.fr (Y.D.)

**Keywords:** ncRNA, pregnancy, low-risk pregnancy, biomarkers, next-generation sequencing, miscarriage, pre-eclampsia, gestational diabetes mellitus, intra-uterine growth restriction, preterm birth

## Abstract

Pregnancy-related complications (PRC) impact maternal and fetal morbidity and mortality and place a huge burden on healthcare systems. Thus, effective diagnostic screening strategies are crucial. Currently, national and international guidelines define patients at low risk of PRC exclusively based on their history, thus excluding the possibility of identifying patients with de novo risk (patients without a history of disease), which represents most women. In this setting, previous studies have underlined the potential contribution of non-coding RNAs (ncRNAs) to detect patients at risk of PRC. However, placenta biopsies or cord blood samples are required, which are not simple procedures. Our review explores the potential of ncRNAs in biofluids (fluids that are excreted, secreted, or developed because of a physiological or pathological process) as biomarkers for identifying patients with low-risk pregnancies. Beyond the regulatory roles of ncRNAs in placental development and vascular remodeling, we investigated their specific expressions in biofluids to determine favorable pregnancy outcomes as well as the most frequent pathologies of pregnant women. We report distinct ncRNA panels associated with PRC based on omics technologies and subsequently define patients at low risk. We present a comprehensive analysis of ncRNA expression in biofluids, including those using next-generation sequencing, shedding light on their predictive value in clinical practice. In conclusion, this paper underscores the emerging significance of ncRNAs in biofluids as promising biomarkers for risk stratification in PRC. The investigation of ncRNA expression patterns and their potential clinical applications is of diagnostic, prognostic, and theragnostic value and paves the way for innovative approaches to improve prenatal care and maternal and fetal outcomes.

## 1. Introduction

In 2019, the yearly birth rate was estimated at approximately 140 million worldwide [1]. Despite new advances in the management of pregnancy, a significant rate of obstetric complications persists related to the increase in maternal age, the number of pregnancies obtained by Assisted Reproduction Technology, maternal obesity, and smoking [2,3,4,5]. In 2020, maternal mortality, stillbirths, and neonatal mortality were estimated at 0.3 million, 1.9 million, and 2.4 million, respectively [6].

Despite better knowledge of the physiopathology of pregnancy, no biological tool (based on serum, plasma, urine, or saliva samples) has been developed to identify patients at low risk of complications during pregnancy. The French health authorities define the concept of low-risk pregnancy based on the analysis of family, personal, gynecological, and obstetrical history according to a non-exhaustive list; a low-risk pregnancy is defined by the absence of history [7]. In addition, the UK’s National Institute for Clinical Excellence (NICE) includes the Body Mass Index (BMI) as a risk factor [8]. While these parameters are commonly recognized risk factors for complications during pregnancy, they are of low utility to predict their occurrence at an individual level. Therefore, there is a need for new biomarkers to identify pregnant women at low or high risk of complications associated with pregnancy to adapt to management. In this specific setting, cumulative evidence suggests the role of non-coding RNAs (ncRNAs) in the pathogenesis of complications associated with pregnancy [9].

## 2. ncRNAs

ncRNAs represent 98% of the transcriptome and include small non-coding RNAs (sncRNAs) composed of less than 50 nucleotides, and long non-coding RNAs (lncRNAs) composed of more than 200 nucleotides. sncRNAs include microRNAs (miRNAs), Piwi-interacting RNAs (piRNAs), transfer RNAs (tRNAs), small nucleolar RNAs (snoRNAs), and small interfering RNAs (siRNAs) [10,11]. lncRNAs include intergenic RNAs (lincRNAs), specific circular RNAs (circRNAs), and ribosomal RNAs (rRNAs). The biological functions of ncRNAs encompass the regulation of gene expression at the transcriptional and translational levels, the guidance of DNA synthesis or gene rearrangement, and the protection of the genome against foreign nucleic acids.

Several studies have highlighted the potential of ncRNAs as biomarkers to diagnose diseases [12,13,14], predict events or complications, and even serve as theragnostic biomarkers [15,16,17]. However, little attention has been paid to biofluid (i.e., a fluid that is excreted, secreted, or developed because of a physiological or pathological process) expression of ncRNAs in the specific setting of pregnancy. The aim of this work was to review the potential contribution of ncRNAs to identify women at low risk of complications associated with pregnancy.

## 3. ncRNAs and Human Placenta

The placenta is an interface between the mother and the fetus and serves as a selective barrier [18,19]. This cellular interface is composed of different subtypes of trophoblast cells of fetal origin (mainly the syncytiotrophoblast) involving numerous ncRNAs, mainly those implicated in angiogenesis (angiomiRs) (Figure 1). The syncytiotrophoblast produces numerous extracellular vesicles containing proteins and ncRNAs that can be released into the maternal circulation (Figure 2). Previous studies have been published on the expression of ncRNAs in placental tissue, mainly focusing on miRNAs, lncRNAs, and circRNAs [20,21]. Numerous reviews have demonstrated the involvement of miRNAs in implantation [22], placental development and function [20], as well as in pathologies associated with pregnancy [23,24,25]. Moreover, it has been reported that miRNAs are sequentially expressed throughout the trimesters of pregnancy, which explains discrepancies between series when ncRNA expression is only assessed once during pregnancy [26,27]. Similar results have been reported for lncRNAs [28,29] and circRNAs [30,31,32,33] in pathologies associated with pregnancy. However, the main limit of placental ncRNA evaluation is the difficulty and risks associated with obtaining tissue samples. Nevertheless, as ncRNAs are highly stable and sequentially expressed, they can be evaluated in peripheral maternal blood [34].

## 4. Pathologies Associated with Pregnancy

### 4.1. Early and Late Miscarriages

Miscarriage is defined as the spontaneous loss of a pregnancy. Around 15% of pregnancies result in an early miscarriage (i.e., <14 weeks of gestation (WG)), and about 1% in a late miscarriage (14–22 WG) [35]. Early miscarriage is caused by a wide range of factors with difficulties in identifying the etiology [36,37]. The most common causes of early miscarriages are chromosomal abnormalities, defective placental development, and maternal disease conditions [36]. The risk of miscarriage per cycle increases with maternal age, ranging from 12% at 25 years to 50% at 42 years [35]. Other risk factors are paternal age, a BMI greater than 25 kg/m^2^, excessive coffee drinking, smoking, alcohol consumption, exposure to magnetic fields and ionizing irradiation, a history of abortion, some fertility disorders, and impaired ovarian reserve.

Although recurrent miscarriages are frequently observed, limited data about ncRNAs in biofluids are available for de novo early miscarriage despite its high incidence [38]. Hosseini et al. evaluated the predictive value of maternal plasma miRNAs for the risk of miscarriage by comparing 16 patients with miscarriage between 6 and 8 WG with eight patients who had abortions at similar terms. They showed that let-7c and miRNA-122 were upregulated and miRNA-135a downregulated in the miscarriage group [21]. Hong et al. reported a decrease in blood miRNA-378a-3p regulating the expression of Caspase-3 and cl-Caspase-3 proteins influencing cell apoptosis in the decidual tissue [39]. Cui et al. reported an overexpression of serum miRNA-371a-5p and down expression of miRNA-206 [40]. Recently, Hromadnikova et al. proposed a predictive model for miscarriage based on the blood expression of eight miRNAs (upregulation of miRNA-1-3p, miRNA-16-5p, miRNA-17-5p, miRNA-26a-5p, miRNA-146a-5p, and miRNA-181a-5p and downregulation of miRNA-130b-3p and miRNA-195-5p) identifying patients at risk in 80.52% of cases [41]. All the above studies are summarized in Table 1. To our knowledge, no data exist about the expression of piRNAs, circRNAs, tRNAs, snoRNAs, or lncRNAs in plasma, serum, urine, or saliva in the context of miscarriage. Moreover, no study has evaluated the expression of ncRNAs in the context of miscarriage using next-generation sequencing (NGS).

### 4.2. Hypertension and Pre-Eclampsia

Recent French guidelines define hypertension during pregnancy as systolic blood pressure (SBP) > 140 mm Hg and/or diastolic blood pressure (DBP) > 90 mm Hg. Chronic hypertension (i.e., hypertension diagnosed before 20 WG) affects fewer than 1% of patients [42]. Gestational hypertension is defined as hypertension diagnosed over 20 WG. The meta-analysis by Li et al. indicated an increased risk of peri- and neonatal death, congenital malformations, intrauterine growth restriction (IUGR), and preterm birth in patients with gestational hypertension [43].

Pre-eclampsia (PE) is defined as a persistent maternal SBP > 140 mm Hg and/or DBP > 90 mm Hg spaced 6 h apart, with proteinuria (≥300 mg protein in a 24 h urine sample or ≥1+ by dipstick) after 20 WG [44]. PE affects 5% of pregnancies, varying with ethnicity, maternal medical history, and socio-economic conditions. PE results from abnormal placentation, causing insufficient uteroplacental blood perfusion and ischemia [45,46]. Severe PE is characterized by an increase in SBP exceeding 160 mm Hg and/or DBP exceeding 110 mm Hg spaced 6 h apart with either mild proteinuria or mild hypertension plus severe proteinuria (≥2 g/24 h or ≥2+ by dipstick) [44].

#### 4.2.1. Expression of miRNAs in Biofluids and PE

The increased expression of angiomiRNAs [47,48] suggests that angiogenesis plays a role in the physiopathology of PE. Furthermore, angiomiRNAs are differentially expressed in placental tissue and maternal blood in PE, IUGR, and gestational diabetes mellitus (GDM) compared with normal pregnancies [49].

In fifteen women with PE (seven mild and eight severe) compared with seven women with a normal pregnancy, Jairajpuri et al. reported that seven miRNAs were differentially expressed in the PE patients: miRNA-215, miRNA-155, miRNA-650, miRNA-210, miRNA-21 were upregulated, and miRNA-18a, miRNA-19b1 were downregulated. Moreover, four miRNAs (upregulated miRNA-518b and miRNA-29a, and downregulated miRNA-144, miRNA-15b) distinguished severe PE from mild PE [50].

In a meta-analysis, Yin et al. reported that the diagnostic values of circulating miRNAs for PE had a sensitivity of 0.88 (95% CI: 0.80–0.93), a specificity of 0.87 (95% CI: 0.78–0.92), and a likelihood ratio of 50.24 (95% CI: 21.28–118.62) [51]. However, the combined sensitivity, specificity, and likelihood ratio of circulating microRNAs for predicting PE in asymptomatic patients was only 0.61 (95% CI: 0.55–0.68), 0.78 (95% CI: 0.72–0.83), and 5.7 (95% CI: 3.7–8.7). In a review, Tsochandaridis et al. emphasized that miRNAs were mainly analyzed from placental samples and were not sufficiently sensitive or specific for clinical diagnosis of early PE [52]. Among these miRNAs, miRNA-144 appears to be an important regulator of ischemia and hypoxia, and miRNA-210 is induced by hypoxia and is regulated by the transcriptional factor HIF-1 (hypoxia-inducible factor) and NF-κB [53]. Additionally, another study showed that miRNA-210 inhibits trophoblast invasion [54]. The authors suggested that these miRNAs could be useful biomarkers for the diagnosis of PE. Additionally, miRNA-210 was found to be significantly increased in the plasma of women with PE [54]. Recently, Mavreli et al. evaluated the expression profile of certain miRNAs in the first-trimester maternal plasma of women who subsequently developed PE compared to uncomplicated pregnancies [55]. More interestingly, compared to studies using micro-arrays or qRT-PCR focusing on a small number of miRNAs selected on predefined criteria, NGS technology allows for the analysis of the entire miRNAome without predefined miRNA selection. In a small series of 5 patients with PE and 5 controls, and subsequently confirmed in an independent cohort of 12 PE cases and 12 controls, Mavreli et al. showed that miRNA-23b-5p and miRNA-99b-5p were downregulated (>1.5-fold) in PE pregnancies compared with controls, and could thus be potential non-invasive biomarkers of PE [55]. We focused on series using NGS which allowed extensive evaluation of ncRNAs (Table 2).

#### 4.2.2. Expression of circRNAs in Biofluids and PE

In a case–control study before 20 WG involving five patients with PE and five controls matched by age and term, Min Jiang et al. evaluated the expression of circRNAs in maternal blood. In the preliminary phase, 2178 circRNAs were differentially expressed between the groups, including 884 downregulated and 1294 upregulated in the PE group. Two circRNAs (circ-0004904 and circ-0001855) were significantly upregulated in the patients with PE [58]. In another case–control study including 41 patients with PE and 41 matched controls, Zhang et al. evaluated the levels of circ-101222 in red blood cells between 8 and 20 WG [59]. Patients with PE had higher levels of circ-101222: sensitivity 0.70, specificity 0.80, and an AUC of 0.87 (95% CI: 0.81–0.92) for predicting PE. Similarly, Zhang et al. reported lower circulating levels of circ-CRAMP1L in patients with PE (2.66 ± 0.82) than in controls (3.95 ± 0.67, *p* < 0.001) with AUC of 0.81 [33]. All the above studies are summarized in Table 3.

No data exist about the expression of circRNAs using NGS in urine or saliva in the context of PE.

#### 4.2.3. Expression of lncRNAs in Biofluids and PE

Sun et al. showed that whole blood levels of lncRNA BC030099 differentiated women with PE from controls (AUC = 0.71) [60]. Luo et al. suggested that the levels of three lncRNAs in the serum (AF085938, G36948, and AK002210) could also serve as diagnostic biomarkers of PE (AUC = 0.76, 0.79, and 0.75, respectively) [61]. Daï et al. identified seven lncRNAs (NR-002187, ENST00000398554, ENST00000586560, TCONS_00008014, ENST00000546789, ENST00000610270, and ENST00000527727) predictive of hypertension and PE but with AUCs between 0.6 and 0.7 [62]. Na Dong et al. reported that lncRNA MIR193BHG was upregulated in the serum of patients with PE [63]. Luo et al. showed that NR-027457 and AF085938 were upregulated, while G36948 and AK002210 were downregulated in PE [61]. Finally, Abdelazim et al. reported that serum MALAT-1 was downregulated while HOTAIR was unaltered in PE. Furthermore, MALAT-1 was reduced in severe PE compared with mild PE and was correlated with gestational age (rho = −0.328) and albuminuria (r = 0.312) [64]. All the above studies are summarized in Table 4.

No data exist about the urine or saliva expression of lncRNAs in the context of PE and eclampsia. Finally, no NGS studies are available evaluating lncRNAs in PE.

To our knowledge, no data exist about the expression of piRNA, tRNA, or snoRNA in blood, serum, urine, or saliva in the context of PE and eclampsia.

### 4.3. Intrauterine Growth Restriction (IUGR)

There are two pathways to low birth weight, preterm birth (PTB) and IUGR, resulting in a small baby for gestational age (SGA). The global incidence of SGA was estimated at 23.4 million in 2020 and is associated with newborn morbidity and mortality [65].

#### 4.3.1. Expression of miRNAs in Biofluids and IUGR

In a recent review, Ali et al. reported that miRNAs were involved in IUGR [66]. While most of the studies analyzed miRNAs in the placenta, some explored the blood. Kochhar et al. also carried out a literature review of miRNAs and IUGR and found that only three of the 21 studies analyzed miRNAs in maternal blood [67]. Among these studies, Hromadnikova et al. evaluated the interest of miRNAs associated with cardiovascular diseases for the early prediction of IUGR in the absence of PE in maternal blood at 10-13 WG. The upregulation of miRNA-1-3p, miRNA-20b-5p, miRNA-126-3p, miRNA-130b-3p, and miRNA-499a-5p was observed in cases of IUGR. The combination of four miRNAs (miRNA-1-3p, miRNA-20a-5p, miRNA-146a-5p, and miRNA-181a-5p) was able to identify 75.68% of pregnancies with IUGR, 4 times more than using the Fetal Medicine Foundation criteria [68]. Tagliaferri et al. evaluated the expression of miRNAs in maternal plasma in women with IUGR vs. control cases. Four miRNAs were identified as possible candidates for the diagnosis of IUGR before 32 WG (miRNA-16-5p, miRNA-103-3p, miRNA-107-3p, and miRNA-27b-3p) [69]. More interesting is the study of Pei et al. that collected maternal blood from 970 women between 10-11 WG, 20-21 WG, and 33-34 WG, showing a progressive elevation of miRNA-590-3p associated with the risk of IUGR, adjusting for maternal age, BMI, and parity [70]. Rodosthenous et al. found higher levels of miRNA-20b-5p, miRNA-942-5p, miRNA-324-3p, miRNA-223-5p, and miRNA-127-3p in maternal serum associated with a greater risk of IUGR [71]. The most pertinent studies are summarized in Table 5.

#### 4.3.2. Expression of lncRNAs in Biofluids and IUGR

Dai et al. evaluated the serum expression of lncRNAs in five patients with IUGR and five controls. Serum levels of ENST00000527727 and ENST00000415029 were associated with a risk of IUGR [62] (Table 6).

One study by Terstappen et al. [72] evaluated the expression of lincRNA RP5-855F14.1 using NGS on umbilical cord vein endothelial cells in a series of 19 patients showing its upregulation in patients with IUGR (Table 6).

No data are available about piRNA, tRNA, circRNA or snoRNA expression in biofluids in IUGR.

### 4.4. Gestational Diabetes Mellitus (GDM)

A lack of pancreatic islet compensation during pregnancy leads to GDM, which increases the risk of type 2 diabetes. Between 24% and 62% of people with diabetes are unaware that they have the disease, are undiagnosed, and untreated [73], suggesting a significant gap in current diagnostic practices. GDM is associated with increased risks of PE, obstetrical intervention, large-for-gestational-age neonates, shoulder dystocia, birth trauma, and neonatal hypoglycemia [74]. Screening and treatment for GDM at 24 to 28 WG are now recommended [75,76]. Women with pregnancies complicated by early (<20 WG) hyperglycemia showed accelerated fetal growth by 24 to 28 WG [77] and had greater perinatal mortality than women who received a diagnosis of GDM later in pregnancy [78]. Furthermore, a linear relationship has been shown between fasting glucose levels in early pregnancy and adverse pregnancy outcomes [79,80].

#### 4.4.1. Expression of miRNAs in Biofluids and GDM

Relatively few data are available about miRNA expression in GDM [81,82,83]. Among the miRNAs most consistently upregulated before 20 WG are miRNA-16-5p, miRNA-17-5p, miRNA-223, miRNA-210-3p, miRNA-342-3p, and miRNA-20a -5p, while miRNA-222 is downregulated. In women at 16 WG, high levels of miRNA-155-5p and miRNA-21-3p were associated with a higher risk of GDM. In pregnant women followed until 28 WG, levels of miRNA-16-5p, miRNA-17-5p, and miRNA-20a-5p were correlated with insulin resistance [84]. Additionally, among GDM patients, women with high levels of miRNA-330-3p had an aggressive diabetic phenotype [85]. However, no miRNA signature has been found in patients with GDM to identify those at risk of complications during pregnancy [86].

In a literature review, Vasu et al. reported miRNAs in the phase of prediabetes and confirmed diabetes [81]. These authors reported that miRNAs are differentially expressed in patients at risk of GDM. In a meta-analysis including 12 studies involving 1768 patients, Lazarus et al. reported that the sensitivity and specificity of miRNAs for diagnosing GDM were 74.5% (95% CI: 63.7–82.9) and 84.1% (95% CI: 76.8–89.3), respectively [87]. The AUC was 0.869 (95% CI: 0.818–0.907). However, the authors underlined limitations linked to the heterogeneity of the studies. A literature review of 16 studies involving 1355 patients (*n* = 674 with GDM, *n* = 681 controls), Lewis et al. showed that 135 miRNAs were associated with GDM, including 8 described in at least two studies (miRNA-16-5p, miRNA -17-5p, miRNA-20a-5p, miRNA-29a-3p, miRNA-195-5p, miRNA-222-3p, miRNA-210-3p, and miRNA-342-3p) [88]. These results suggest that miRNA levels vary depending on the term of pregnancy when GDM develops, fetal sex, and BMI both before and during pregnancy. The most pertinent studies are summarized in Table 7.

No data exist about the expression of miRNAs in urine or saliva in the context of GDM. No studies to date have evaluated miRNA expression in GDM using NGS.

#### 4.4.2. Expression of circRNAs in Biofluids and GDM

Zhang et al. showed that circRNAs are differentially expressed in serum during the second and third trimesters of pregnancy [89]. Focusing on insulin resistance, these authors reported circRNAs expressed in prediabetes, type 1 and 2 diabetes and in GDM. However, the data are insufficient to clarify the contribution of circRNAs in the diagnosis and prognosis of GDM.

No data exist about the expression of circRNAs in urine or saliva or using NGS.

#### 4.4.3. Expression of lncRNAs in Biofluids and GDM

Li Yuanyuan et al. evaluated the differential expression of lncRNAs in the blood of patients with GDM and in a control group [90]. Using lncRNA microarrays, 1098 lncRNAs were differentially expressed (609 upregulated, 489 downregulated). Several lncRNAs have been shown to play an important role in insulin resistance (ERMP1, TSPAN32, and MRPL38 forming a co-expression network with TPH1) and in the development of GDM. Furthermore, lncRNA RPL13P5 forms a co-expression network with the TSC2 gene via PI3K-AKT and insulin signaling pathways, which are involved in the process of insulin resistance in GDM. Additional studies demonstrated the involvement of other lncRNAs in GDM: lncRNA-MEG8 [91], lncRNA GAS5 [92], lncRNA MEG3 [93], and the lncRNA-UCA1/miRNA-138 axis [94]. Similarly, Yisheng Zhang et al. found that the expression level of lncRNA MALAT1 was higher in GDM (*p* = 0.007), and that the expression of IncRNA MALAT1 was correlated with lncRNA p21 (r = 0.333, *p* = 0.018) and lncRNA H19 (r = 0.314, *p* = 0.030) [95] (Table 8). Ruifen Su et al. noted that HOTAIR was both of diagnostic value for GDM and was positively correlated with BMI, fasting blood glucose, and 1 h and 2 h blood glucose [96] (Table 8). Finally, Jingjun Li et al. observed that SNHG17 was downregulated in patients with GDM and allowed for the diagnosis of GDM 4 weeks before diagnosis by the standard method [97] (Table 8).

Only one study used NGS to evaluate the expression of ncRNAs in GMD. In a series of 60 patients, Yan et al. [30] reported a differential expression of various circRNAs in GDM (circRNA.17543 circRNA.17415, circRNA.14701 circRNA.32231, circRNA.9695, circRNA.47552, circRNA.20697, circRNA.746, circRNA.19482, circRNA.40427, circRNA.18066, circRNA.1030, circRNA.5673, circRNA.5333, circRNA.2469, circRNA.2776, circRNA.42159, circRNA.40647, circRNA.23658, circRNA.33922, circRNA.39475, circRNA.5092, circRNA.20159, circRNA.11829, circRNA.17541, circRNA.2789, circRNA.27805, circRNA.28734, circRNA.28798, circRNA.21872, circRNA.16180, circRNA.27819, circRNA.1898, circRNA.35218, circRNA.6929, circRNA.37134, circRNA.36044, circRNA.22054, circRNA.34162, and circRNA.22055).

No data exist about the expression of lncRNA in urine or saliva. No data exist about the expression of piRNA, tRNA, or snoRNA in plasma, serum, or saliva.

### 4.5. Preterm Birth (PTB)

PTB, described as birth < 37 WG [98], complicates 5–18% of pregnancies [99]. It remains the leading cause of mortality in children under 5 years and is responsible for about 1 million deaths annually [100]. Global estimates from 2020 suggest that 13.4 million live births were preterm, with rates over the past decade remaining stable [65].

#### 4.5.1. Expression of miRNAs in Biofluids and PTB

Using peripheral blood or amniotic fluid, Yang et al. [101] showed that miRNA-302b, miRNA-548, and miRNA-1253 are downregulated, while miRNA-223 was upregulated in cases of PTB [102] and could predict the risk of PTB and premature rupture of membranes (PROM) [103,104,105].

In 2017, Gray et al. reported an increase in plasma miRNA-223 and a decrease in miRNA-302b, miRNA-1253, and miRNA-548 in a population of seven women who had PTB and nine controls [102]. Menon et al. evaluated the plasma expression of miRNAs in 20 women with PTB during the three trimesters of pregnancy. A total of 167 and 153 miRNAs were found to be differentially expressed (*p* < 0.05) according to gestational age for term pregnancies and PTB, with variations depending on the term, highlighting the interest in a sequential analysis of miRNAs during pregnancy [106].

Illarionov et al. carried out a prospective study of miRNAs in plasma during the first and second trimesters in pregnant women at high risk of PTB (13 cases and 11 controls) [107]. Plasma blood samples at 9–13 WG and 22–24 WG were available for the study group. Using high-throughput sequencing technology, differences were found during the first trimester compared with control (pregnant women without risk of PTB) in the levels of 15 miRNAs: 3 upregulated (miRNA-122-5p, miRNA-34a-5p, and miRNA-34c-5p) and 12 downregulated (miRNA-487b-3p, miRNA-493-3p, miRNA-432-5p, miRNA-323b-3p, miRNA-369-3p, miRNA-134-5p, miRNA-431-5p, miRNA-485-5p, miRNA-382-5p, miRNA-369-5p, miRNA-485-3p, and miRNA-127-3p) (log2(FC) ≥ 1.5; FDR ≤ 0.05).

Mavreli et al., in a case–control study of five patients, reported a decrease in miRNA-23b-5p and miRNA-125a-3p and an increase in miRNA-4732-5p in PTB [108]. Cook et al. collected plasma between 12 and 22 WG in women with PTB and/or cervical shortening (6 controls, 13 PTB, and 24 short cervix) to evaluate miRNA expression. Subsequently, a validation study (96 controls, 14 PTB, and 21 with cervical shortening at <20 WG) found nine miRNAs (let-7a-5p, miRNA-374a-5p, miRNA-15b-5p, miRNA-19b-3p, miRNA-23a-3p, miRNA-93-5p, miRNA-150-5p, miRNA-185-5p, and miRNA-191-5p) differentially expressed (*p*  <  0.001) in cases of PTB or cervical shortening. miRNA-150-5p had the highest predictive value of PTB (AUC  =  0.8725) and cervical shortening (AUC = 0.8514) [109].

More interesting is the study by Hromadnikova et al. evaluating plasma miRNAs associated with cardiovascular diseases between 10 and 13 WG to predict PBT or PROM in the absence of other pregnancy-related complications in 6440 patients including 41 with PTB and 65 with PROM [68]. Downregulation of miRNA-16-5p, miRNA-20b-5p, miRNA-21-5p, miRNA-24-3p, miRNA-26a-5p, miRNA-92a-3p, miRNA-126-3p, miRNA- 133a-3p, miRNA-145-5p, miRNA-146a-5p, miRNA-155-5p, miRNA-210-3p, miRNA-221-3p, and miRNA-342-3p was observed in cases of PTB < 37 WG. Only miRNA-210-3p was downregulated in PTB < 34 WG. Decreased miRNA-24-3p, miRNA-92a-3p, miRNA-155-5p, and miRNA-210-3p were a common feature of PTB and PROM.

Winger et al. evaluated 45 miRNAs in blood collected between 6 and 12 WG from 139 women with term delivery (>37 WG) and 18 PTB (<35 WG). The AUC for predicting PTB was 0.80 (95% CI: 0.69–0.88; *p* = 0.0001), with a sensitivity of 0.89 but a specificity of 0.71 [110]. All the abode studies are summarized in Table 9.

No data are available about the expression of miRNAs in urine or saliva in PTB or PROM. No studies have used NGS to evaluate ncRNAs in GDM.

#### 4.5.2. Expression of circRNAs in Biofluids and PTB

In a preliminary study, Ran et al. evaluated the expression of circRNAs in maternal and fetal samples from preterm and term pregnancies, including maternal plasma, maternal monocytes, myometrium, chorion, placenta, and cord blood. They identified 261 circRNAs differentially expressed in PTB. Among these, 26 circRNAs were upregulated, and the others were downregulated. The most significant was SMARCA5_0005 (log2FC = −8.88, adj. *p* < 0.001) [111].

No data are available about the expression of circRNAs in urine or saliva in PTB or PROM or using NGS.

#### 4.5.3. Expression of lncRNAs in Biofluids and PTB

Zhou et al. reported that two lncRNA transcripts in maternal blood from the third trimester were differentially correlated with PTB: LINC00094 (r = 0.196) and LINC00870 (r = −0.303) [112].

No data are available about the expression of lncRNAs in urine or saliva in PTB or PROM.

No data exist about the expression of piRNA, tRNA, or snoRNA in plasma, serum, or saliva in PTB. No studies have used the NGS technique to date.

## 5. Perspectives and Conclusions

The present review of ncRNA expression in biofluids confirms their contribution to the understanding of the physiopathology of the principal etiologies of PRC. However, few data are available about miscarriage despite its high incidence, especially for early miscarriage. Moreover, it is not possible to conclude about the usefulness of analyzing ncRNA expression in routine clinical practice. On the other hand, several reports, including a meta-analysis, support the contribution of ncRNAs in the diagnosis and prognosis of PE, justifying further studies to evaluate the contribution of miRNAs, circRNAs, and lncRNAs to manage patients at risk of hypertension. IUGR represents a crucial issue not only for the newborn and the family but also because of the substantial subsequent burden on healthcare resources. Therefore, ncRNA screening (mainly miRNAs) in biofluids should be introduced in routine clinical practice to detect this severe pathology as early as possible. Because of the increase in obesity in developed countries, GDM represents another major issue. Despite several national and international guidelines, GDM is often detected late, resulting in increased morbidity for the pregnant woman, the fetus, and even the future child. In this setting, it has been demonstrated that miRNA expression can diagnose the prediabetes phase, underlining its potential role in clinical practice. Finally, the incidence of PTB, often the consequence of the above-mentioned pathologies, could be decreased by better screening of ncRNAs in biofluids, especially in saliva.

Despite the abundant literature, it is not possible to identify patients at low risk of complications. This can be explained by various factors. First, the retrospective nature of most studies with small sample sizes means that it is not possible to draw clear conclusions. Moreover, most studies failed to evaluate the ncRNAs sequentially throughout pregnancy, thus limiting their use as diagnostic and prognostic biomarkers of pregnancy-associated pathologies. Second, the techniques used to evaluate ncRNAs that are widely affordable, mainly based on micro-array or qRT-PCR, can only determine a limited number of ncRNAs. Conversely, techniques based on combined NGS and artificial intelligence with machine learning have demonstrated diagnostic relevance in other diseases [113]. Indeed, in the context of endometriosis, it has been possible to establish a saliva miRNA signature, overcoming previous diagnostic barriers. In addition, among the 2600 miRNAs of the miRNAome, some previously unreported miRNAs have been shown to be involved in new pathophysiological pathways with potential therapeutic consequences [114]. Third, another crucial limitation of previous studies is the characteristics of the control group. Indeed, pathologies associated with pregnancy involve a complex array of risk factors. This is the case, for example, for hypertension and PE, which are associated with the risk of IUGR, PROM, as well as spontaneous and induced PTB. In the same way, GDM is associated with obesity, hypertension, and induced PTB. Therefore, as previously underlined [68], there is a need to include patients without any pathology associated with pregnancy in the control group to identify patients at low risk of complications associated with pregnancy. Finally, no consensus exists about which biofluid to use. However, a saliva sample emerges as an ideal biofluid candidate as it is highly reproducible and stable [115] and easy to sample and transport for geographically isolated patients.

All these considerations should be considered to design a prospective trial to identify a population of women at low risk of complications associated with pregnancy. This ties in with a societal trend towards a decrease in the medical management of pregnancy and an increasing desire for home births: the identification of patients at low risk of complications associated with pregnancy could help select patients with a high chance of normal physiologic labor and birth. Finally, identifying a population at low risk of complications will contribute to a better use of healthcare resource consumption.

## Figures and Tables

**Figure 1 diagnostics-14-00729-f001:**
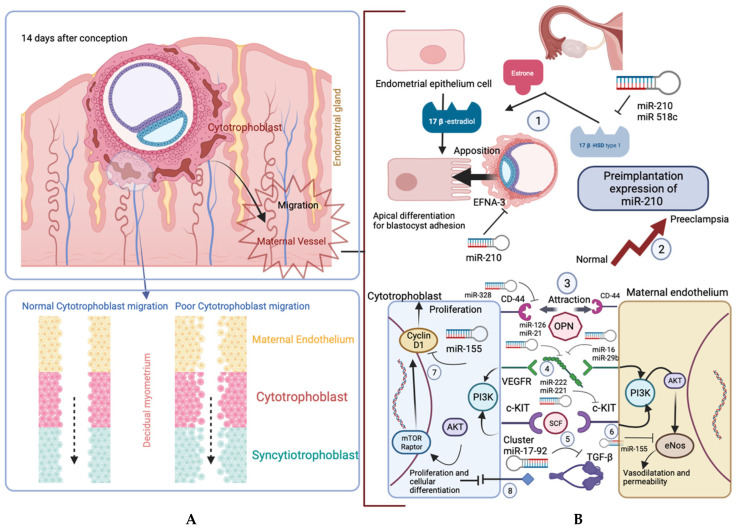
Main pathways and roles of angiomiRs in placentation. The top panel of section (**A**) shows key events following the blastocyst’s attachment to the endometrial epithelium. The bottom panel shows cytotrophoblast invasion, which involves migration and proliferation. This process increases vascular capacity, and poor migration can lead to reduced vessel transformation, causing elevated blood pressure and decreased blood flow. Section (**B**) illustrates some placental angiomiRs associated with conditions like pre-eclampsia (PE), intrauterine growth restriction (IUGR), and/or gestational diabetes mellitus (GDM). 1. Before implantation, the enzyme 17*β* HSD 1 transforms estrone to estradiol in the corpus luteum, inducing endometrial epithelial differentiation for blastocyst attachment. 2. mliR-210 and miR-518c target and regulate 17*β* HSD 1. 3. miR-210 regulates EFNA3, expressed on trophectoderm cells near the inner cell mass of the blastocyst, guiding its location. Examples illustrate potential concerns about miR-210 overexpression, as seen in PE. 4. We show the role of miR-328 on the hyaluronate receptor CD44, involved in the migration of cytotrophoblast and endothelial cells in maternal vessels, responding to the gradient of the OPN protein released from endometrial glands. 5. The regulation of VEGF species by miR-126 and miR-21, as well as miR-16 and miR-29b on VEGFA, appears crucial for promoting angiogenesis. 6. miR-222 and miR-221 regulate c-Kit, and their overexpression disrupts the balance with its ligand SCF, affecting proliferative signals. 7. miR-155 downregulating eNOS is observed, potentially impairing vasodilation and permeability. Additionally, miR-155 targets cyclin D1, leading to proliferation alteration. 8. The cluster miR-17-92 regulating TGF*β*, a key growth factor in cytotrophoblast proliferation and differentiation, is noted.

**Figure 2 diagnostics-14-00729-f002:**
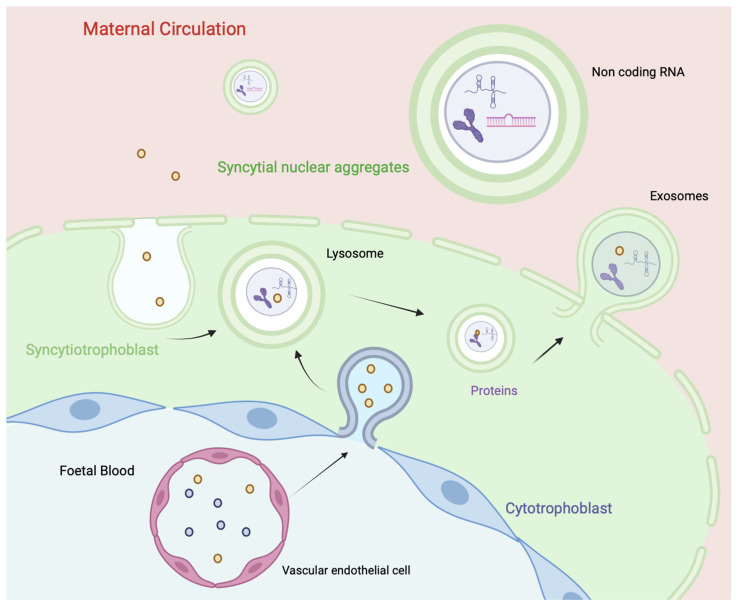
Pathway of genetic material, including ncRNAs, from the fetal to the maternal circulation, including the pathway of fetal vascular endothelial cells through the cytotrophoblast and syncytiotrophoblast to the maternal circulation.

**Table 1 diagnostics-14-00729-t001:** Summary of studies evaluating ncRNAs in miscarriage.

Author	Year	Number of Patients	Type of ncRNA	Type of Sample
Hosseini et al. [21]	2018	16	Upregulated: Let-7c, miRNA-122 Downregulated: miRNA-135a	Plasma
Hong et al. [39]	2018	100	Downregulated: miRNA-378a-3p	Decidua
Cui et al. [40]	2021	68	Upregulated: miRNA-371a-5p Downregulated: miR-206	Serum
Hromadnikova et al. [41]	2023	181	Upregulated: miRNA-1-3p, miRNA-16-5p, miRNA-17-5p, miRNA-26a-5p, miRNA-146a-5p, miRNA-181a-5p Downregulated: miRNA-130b-3p, miRNA-195-5p	Peripheral venous blood

**Table 2 diagnostics-14-00729-t002:** Summary of NGS-based studies for ncRNA sequencing in PE.

Author	Year	Number of Patients	Type of ncRNA	Type of Sample
Timofeeva et al. [56]	2017	54	Placenta downregulated miRNA-532-5p, miRNA-423-5p, miRNA-127-3p, miRNA-539-5p, miRNA-519a-3p, and miRNA-629-5p and let-7c-5p Plasma upregulated miRNA-423-5p, miRNA-519a-3p, and miRNA-629-5p and let-7c-5p	Placenta Plasma
Mavreli et al. [55]	2020	10	miRNA-23b-5p miRNA-99b-5p downregulated	Plasma
Chamberlain et al. [57]	2023	23	Not enough materiel	Serum

**Table 3 diagnostics-14-00729-t003:** Summary of studies evaluating circRNAs in PE.

Author	Year	Number of Patients	Type of circRNA	Type of Sample
Zhang et al. [59]	2016	82	Upregulated: circ-101222	Red blood cells
Min Jiang et al. [58]	2018	10	Significantly upregulated: circ-0004904, circ-0001855	Maternal blood
Zhang et al. [33]	2020	128	Downregulated: circ-CRAMP1L	Plasma

**Table 4 diagnostics-14-00729-t004:** Summary of studies evaluating lncRNAs in PE.

Author	Year	Number of Patients	Type of lncRNA	Type of Sample
Sun et al. [60]	2019	72	Upregulated: BC030099	Maternal blood
Luo et al. [61]	2019	162	Upregulated: AF085938 Downregulated: G36948 and AK002210	Serum
Daï et al. [62]	2021	10	NR-002187, ENST00000398554, ENST00000586560, TCONS_00008014, ENST00000546789, ENST00000610270, ENST00000527727	Serum
Na Dong et al. [63]	2022	166	Upregulated: MIR193BHG	Serum
Abdelazim et al. [64]	2022	160	Downregulated: MALAT-1	Serum

**Table 5 diagnostics-14-00729-t005:** Summary of studies evaluating miRNAs in IUGR.

Author	Year	Number of Patients	Type of miRNA	Type of Sample
Rodosthenous et al. [71]	2017	100	Upregulated: miRNA-20b-5p, miRNA-942-5p, miRNA-324-3p, miRNA-223-5p, miRNA-127-3p	Maternal serum
Tagliaferri et al. [69]	2021	77	miRNA-16-5p, miRNA-103-3p, miRNA-107-3p, miRNA-27b-3p	Maternal plasma
Hromadnikova et al. [68]	2022	258	Upregulated: miRNA-1-3p, miRNA-20b-5p, miRNA-126-3p, miRNA-130b-3p, and miRNA-499a-5p	Maternal blood
Pei et al. [70]	2022	970	Upregulated: miRNA-590-3p	Maternal blood

**Table 6 diagnostics-14-00729-t006:** Summary of studies evaluating lncRNAs in IUGR.

Author	Year	Number of Patients	Type of lncRNA	Type of Sample
Terstappen et al. [72]	2020	19	Upregulated: lincRNA RP5-855F14.1	Umbilical cord vein endothelial cells
Dai et al. [62]	2021	10	ENST00000527727, ENST00000415029	Maternal serum

**Table 7 diagnostics-14-00729-t007:** Summary of studies evaluating miRNAs in GDM.

Author	Year	Number of Patients	Type of miRNA	Type of Sample
Cao et al. [84]	2017	157	Upregulated: miRNA-16-5p, miRNA-17-5p, miRNA-20a-5p	Maternal plasma
Sebastiani et al. [85]	2017	31	Upregulated: miRNA-330-3p	Maternal plasma
Lewis et al. [88]	2023	1355	135 miRNAs associated with GDM	Maternal plasma/serum

**Table 8 diagnostics-14-00729-t008:** Summary of studies evaluating lncRNAs in GDM.

Author	Year	Number of Patients	Type of lncRNA	Type of Sample
Yisheng Zhang et al. [95]	2017	97	Upregulated: l lncRNA MALAT1	Maternal serum
Ruifen Su et al. [96]	2021	198	Upregulated: l lncRNA HOTAIR	Maternal blood
Jingjun Li et al. [97].	2021	360	Downregulated: lncRNA SNHG17	Maternal blood

**Table 9 diagnostics-14-00729-t009:** Summary of studies evaluating miRNAs in PTB.

Author	Year	Number of Patients	Type of miRNA	Type of Sample
Gray et al. [102]	2017	16	Upregulated: miRNA-223 Downregulated: miRNA-302b, miRNA-548, miRNA-1253	Maternal plasma
Menon et al. [106]	2019	30	A total of 167 and 153 miRNAs were found to be differentially expressed (*p* < 0.05)	Maternal plasma
Cook et al. [109].	2019	43 and then validation with 131 patients	9 miRNAs differentially expressed: let-7a-5p, miRNA-374a-5p, miRNA-15b-5p, miRNA-19b-3p, miRNA-23a-3p, miRNA-93-5p, miRNA-150-5p, miRNA-185-5p, miRNA-191-5p	Maternal plasma
Winger et al. [110]	2020	157	45 miRNAs	Maternal blood
Illarionov et al. [107].	2022	24	Upregulated: miRNA-122-5p, miRNA-34a-5p, miRNA-34c-5p Downregulated: miRNA-487b-3p, miRNA-493-3p, miRNA-432-5p, miRNA-323b-3p, miRNA-369-3p, miRNA-134-5p, miRNA-431-5p, miRNA-485-5p, miRNA-382-5p, miRNA-369-5p, miRNA-485-3p, miRNA-127-3p	Maternal plasma
Mavreli et al. [108]	2022	5	Upregulated: miRNA-4732-5p Downregulated: miRNA-23b-5p, miRNA-125a-3p	Maternal plasma
Hromadnikova et al. [68].	2022	6440 patients including 41 with PTB and 65 with PROM	Downregulated: miRNA-16-5p, miRNA-20b-5p, miRNA-21-5p, miRNA-24-3p, miRNA-26a-5p, miRNA-92a-3p, miRNA-126-3p, miRNA- 133a-3p, miRNA-145-5p, miRNA-146a-5p, miRNA-155-5p, miRNA-210-3p, miRNA-221-3p, miRNA-342-3p.	Maternal plasma

## Data Availability

Data is contained within the article.

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
