# Peer review of "Value of Non-Coding RNA Expression in Biofluids to Identify Patients at Low Risk of Pathologies Associated with Pregnancy"

_diagnostics, 2024, doi:10.3390/diagnostics14070729_

Round 1

Reviewer 1 Report

Comments and Suggestions for Authors

I have advanced training in obstetrics and intermediate training in basic sciences. The topic is interesting, but the article is extremely difficult for a clinician to follow. In fact, the most important pregnancy complications (preeclampsia, fetal growth retardation, fetal death, premature birth) are syndromes whose pathophysiology is not well understood and probably multifactorial. For each of these pathologies, there are articles and books of dozens, even hundreds of pages. I would therefore advise authors to limit their interpretation of results to the associations found, rather than to mechanistics. In fact, authors should highlight associations that have been demonstrated and validated in independent cohorts with the requisite power. 

I have trouble understanding some sentences. The most shocking is this one: "Early and late miscarriages : Miscarriage, defined as the spontaneous loss of a pregnancy, is estimated to 15% of pregnancies end in early miscarriage (<14 weeks of gestation (WG)), and late miscarriage (14-22 WG) concerns about 1% of pregnancies (35). It is considered that intrauterine growth restriction is the main cause, along with prematurity, perinatal morbidity and mortality (36). Maternal age increases the risk of miscarriage per cycle, increasing from 12% to 50% between 25- and 42-years-old (35)."

Firstly, miscarriages occur in over 15% of pregnancies, the main known cause remains chromosomal abnormalities, and fetal growth retardation is by no means a cause of miscarriages, but rather a cause of late fetal death. Secondly, I am not sure what they really meant by those sentences...

Clearly, the authors should concentrate on the "omic" aspect of the paper, and stay away from aspects of pregnancy they haven't mastered.

Comments on the Quality of English Language

I have trouble understanding some sentences.

The most shocking is this one:

Miscarriage, defined as the spontaneous loss of a pregnancy, is estimated to 15% of pregnancies end in early miscarriage (<14 weeks of gestation (WG)), and late miscarriage (14-22 WG) concerns about 1% of pregnancies (35).

I'll discuss about it later.

Author Response

Dear reviewer,

Thank for your kind comments and evaluation of our work.

Comments and Suggestions for Authors

I have advanced training in obstetrics and intermediate training in basic sciences. The topic is interesting, but the article is extremely difficult for a clinician to follow. In fact, the most important pregnancy complications (preeclampsia, fetal growth retardation, fetal death, premature birth) are syndromes whose pathophysiology is not well understood and probably multifactorial. For each of these pathologies, there are articles and books of dozens, even hundreds of pages. I would therefore advise authors to limit their interpretation of results to the associations found, rather than to mechanistics. In fact, authors should highlight associations that have been demonstrated and validated in independent cohorts with the requisite power. 

Response:

First of all, we want to thank the reviewer for the comments. We totally agree that the pathophysiology of the most frequent pregnancy complication is not well understood and probably multifactorial. This explains that our aim was not to focus on one pathology such as preeclampsia, fetal growth restriction or gestational diabetes mellitus, as there are entanglements between the various pathologies. Concerning the literature on the subject so far there are only studies comparing patient with and without one pathology associated with pregnancy. Moreover, to our knowledge no prospective study with power calculation is available. Effectively the second aim of this article was to underline the need for prospective studies to evaluate the contribution of non-coding RNAs in both the diagnosis, the predictive value and potentially the theragnostic value of non-coding RNAs expression in biofluids.

I have trouble understanding some sentences. The most shocking is this one: "Early and late miscarriages: Miscarriage, defined as the spontaneous loss of a pregnancy, is estimated to 15% of pregnancies end in early miscarriage (<14 weeks of gestation (WG)), and late miscarriage (14-22 WG) concerns about 1% of pregnancies (35). It is considered that intrauterine growth restriction is the main cause, along with prematurity, perinatal morbidity and mortality (36). Maternal age increases the risk of miscarriage per cycle, increasing from 12% to 50% between 25- and 42-years-old (35)."

Firstly, miscarriages occur in over 15% of pregnancies, the main known cause remains chromosomal abnormalities, and fetal growth retardation is by no means a cause of miscarriages, but rather a cause of late fetal death. Secondly, I am not sure what they really meant by those sentences...

Clearly, the authors should concentrate on the "omic" aspect of the paper and stay away from aspects of pregnancy they haven't mastered.

Response:

We totally agree the comment and we suggest deleting the sentence “It is considered that intrauterine growth restriction is the main cause, along with prematurity, perinatal morbidity and mortality (36)”.

Moreover, to clarify the text we suggest adding several sentences: “Miscarriage is caused by a wide range of factors with difficulties to identify the aetiology (36, 37). The most common causes of early miscarriages are chromosomal abnormalities, defective placental development, and maternal disease conditions (36).”
As mentioned in the text, in contrast to recurrent miscarriage, little data are available on de novo early miscarriage that is associated with major dissatisfaction of the patient with the risk of stigma. Our objective was not to focus on recurrent miscarriage but to evaluate whether some non-coding RNAs expressed in biofluids could contribute to identify patients at risk. As noted, the data are very scarce.

Concerning the late miscarriage, few data are available to evaluate the contribution of non-coding RNAs to diagnose and to predict the risks. It is important to note that the goal of our article was to identify patients at low risk of pathologies associated with pregnancy, thus excluding patients with known prior complications. This explains the absence of data and the requirement to evaluate since the first trimester of the pregnancy by biology and especially the expression of non-coding RNAs in biofluids patients with a risk of late miscarriage. This is possible only using a prospective study on a large cohort including nulliparous or multiparous women to define those at low risk of pregnancy-associated complications and thus to adapt the management of patients at risk.

Comments on the Quality of English Language

I have trouble understanding some sentences.

The most shocking is this one:

Miscarriage, defined as the spontaneous loss of a pregnancy, is estimated to 15% of pregnancies end in early miscarriage (<14 weeks of gestation (WG)), and late miscarriage (14-22 WG) concerns about 1% of pregnancies (35).

I'll discuss about it later.

Response:

We totally agree the comment and we suggest reformulating the following sentence: “Miscarriage, defined as the spontaneous loss of a pregnancy, is estimated to 15% for early miscarriage (<14 weeks of gestation (WG)), and about 1% for late miscarriage (14-22 WG) (35).”

Moreover, we suggest sending our paper for correction of the English.

Reviewer 2 Report

Comments and Suggestions for Authors

 Dear Authors,

 I would like to express my sincere appreciation for your paper titled "Value of non-coding RNAs expression in biofluids to identify patients at low risk of pathologies associated with pregnancy," sent to the journal "Diagnostic" for peer review. I am thoroughly impressed by the specificity and clarity with which you delve into the complex topic of non-coding RNA in pregnancy.

Your dedication to shedding light on this emerging area of research is commendable, and your insights are invaluable for advancing our understanding of how non-coding RNAs can be utilized to identify patients at low risk of pregnancy-related pathologies.

Here are some changes that I recommend to emprove the scientific value of your paper:

General changes:

·        Please change the paragraph settings. Where is the main text? Change the paragraph and sub-paragraph heading to be more clear.

·        Please add a comprehensive table on all the study cited in the text with the specification of author, year, number of patient, type of molecule, Type of sample and outcomes.

·        Please check all the links to websites in the bibliography because some don’t’ work properly.

·        In the conclusions, please reassume all the findings of your study.

·        The general understanding of English language is good. Hovewer I recommend made a general spell check to reach the level of English of the Journal.

·        The general format of images is of bad resolution. Please upload high resolution images.

Abstract

·        Line 8-9: Please define better  the significance of “de novo risk” in the sentence

·        Line 12: Please define accurately “biofluids”

·        Line 19-20 Please be more accurately in the conclusions of your research, in particular what are the emerging factors and the new sector of study

MAIN TEXT

·        LINE 27: Reference 1: the link is invalid. Please use a more accurate reference for the sentence.

·        Line 33: The sentence is not clear. Define “biological tool”.

·        Line 37: Reference 7: the link is invalid.

·        Line 45 : sncRNAs stands for Small noncoding RNAs. Change it please.

·        Line 46: lncRNAs  stands for long noncoding RNAs . Change it please.

·        Line 78: Figure 1: please upload a more resolution image. Is the image original? If not, remove it! Maybe you can make more simple the signaling pathway (make another image on it should be an option)

·        Line 80-101: is this the legenda of Figure 1? In this case change the format according to the journal layout.

·        Line 102-104: please add an accurate description of the figure. Is the image original? If not, remove it!

·        Line 112: Please change 25kg/m2 in 25kg/m2

·        Line 115: Please define “biofluids” when it is cited the first time in the main text. Please change “Hossein et al” in  “Hosseini et al.”

·        Line 133: reference 41: please specifiy

·        Line 139: reference 44 and 45 not appear related to the sentence

·        Line 159: Please specify if the likelihood ratio is 50 or 24

·        Line 176: Mavreli  et  al  change in “Mavreli  et  al.”

·        Line 181: Please format properly the table and order the cases by date. Please add data on Nagy et al. (57) or remove it. Add an explanation after the table please. Please change “Not enough materiel” in english.

·        Line 209: what (r=- 0.328) stands for?.

·        Line 222. Add the point to et al.

·        Line 394-424: Please add a more specific sentences on the state-of-art of what is discovered now on the topic.

·        Line 423-424: Please add a more specific conclusion on the health care impact of those (potentials) discovers.

Comments on the Quality of English Language

·        The general understanding of English language is good. Hovewer I recommend made a general spell check to reach the level of English of the Journal. 

Author Response

Dear reviewer,

Thank for your kind comments and evaluation of our work.

Comments and Suggestions for Authors

Dear Authors,

 I would like to express my sincere appreciation for your paper titled "Value of non-coding RNAs expression in biofluids to identify patients at low risk of pathologies associated with pregnancy," sent to the journal "Diagnostic" for peer review. I am thoroughly impressed by the specificity and clarity with which you delve into the complex topic of non-coding RNA in pregnancy.

Your dedication to shedding light on this emerging area of research is commendable, and your insights are invaluable for advancing our understanding of how non-coding RNAs can be utilized to identify patients at low risk of pregnancy-related pathologies.

Here are some changes that I recommend to improve the scientific value of your paper:

General changes:

  • Please change the paragraph settings. Where is the main text? Change the paragraph and sub-paragraph heading to be more clear.

Response: We totally agree with the comment and changed the paragraph and sub-paragraphs heading through the text.

  • Please add a comprehensive table on all the study cited in the text with the specification of author, year, number of patients, type of molecule, Type of sample and outcomes.

Response: We added a comprehensive table for important studies cited in the text at the end of each relevant paragraph.

  • Please check all the links to websites in the bibliography because some don’t’ work properly.

Response: We verified the 4 weblinks included in the references.

  • In the conclusions, please reassume all the findings of your study.

Response: We added the following in the text: “Among the various pathology associated with pregnancy, only few data are available on miscarriage despite its high incidence, especially for early miscarriage. Moreover, it is not possible to state on their role in clinical routine. Concerning hypertension and preeclampsia, several reports including meta-analysis support the contribution of non-coding RNAs in the diagnosis and the prognosis of PE justifying further studies to evaluate the contribution of miRNAs, circRNAs and lncRNAs to manage patients with risk of hypertension. IUGR represents a crucial issue with major risk for the newborn, the family but also for the society requiring high healthcare resource consumption. Therefore, it should be necessary to evaluate as soon as possible, the inclusion of non-coding RNAs (mainly miRNAs) in biofluids to detect this severe pathology. Due to the increase of obesity in developed countries, GDM represents another major issue. Despite several national and international guidelines to diagnose GDM, the detection of patient is often late thus associated with subsequent risk of increased morbidity for the pregnant woman, the fetus and even the future child. In this setting, it has been demonstrated that miRNAs expression allows to diagnose the prediabetes phase underlining its potential role in clinical routine. Finally, PTB, often the consequence of the aforementioned pathologies, could be decrease by a better screening thanks to the availability of ncRNAs in biofluids, especially in saliva.”

  • The general understanding of English language is good. Hovewer I recommend made a general spell check to reach the level of English of the Journal.

Response: we suggest sending our paper for correction of the English.

  • The general format of images is of bad resolution. Please upload high resolution images.

Response: We modified the figure 1 to improve the reading.

Abstract

  • Line 8-9: Please define better the significance of “de novo risk” in the sentence

Response: We add a definition in the text “(patient without prior history)”

  • Line 12: Please define accurately “biofluids”.

Response: We add a definition in the text “fluids that can be excreted, secreted or develop as a result of a physiological or pathological pro-cesses”

  • Line 19-20 Please be more accurately in the conclusions of your research, in particular what are the emerging factors and the new sector of study

Response: We modified the sentence in the abstract: “The investigation of ncRNA expression patterns and their potential clinical applications is diagnosis, prognosis and theragnosis value, paves the way for innovative approaches to enhance pre-natal care and improve maternal and foetal outcomes.

MAIN TEXT

  • LINE 27: Reference 1: the link is invalid. Please use a more accurate reference for the sentence.

Response: We verified and the link works.

  • Line 33: The sentence is not clear. Define “biological tool”.

Response: We added a definition in the text: “(serum, plasma, urine or saliva samples)”

  • Line 37: Reference 7: the link is invalid.

Response: Sorry but the link works.

  • Line 45: sncRNAs stands for Small noncoding RNAs. Change it please.

Response: Thanks for the comment. We changed it.

  • Line 46: lncRNAs  stands for long noncoding RNAs . Change it please.

Response: Thanks for the comment. We changed it.

  • Line 78: Figure 1: please upload a more resolution image. Is the image original? If not, remove it! Maybe you can make simpler the signaling pathway (make another image on it should be an option)

Response: As suggested, we modified the figure in a better resolution. Furthermore, the image is original and made by Dr Amelia Favier. It is difficult to simplify such a complex context as we previously did in this initial figure.

  • Line 80-101: is this the legenda of Figure 1? In this case change the format according to the journal layout.

Response: As suggested, we modified the format legenda.

  • Line 102-104: please add an accurate description of the figure. Is the image original? If not, remove it!

Response: We added a clearer version of the legenda of figure 2 “Pathway of genetic material, including ncRNAs, from the foetus circulation to the maternal circulation including the pathway from fetal vascular endothelial cell through cytotrophoblast and syncitiotrophoblast to maternal circulation.”
Furthermore, the image is original and made by Dr Amelia Favier.

  • Line 112: Please change 25kg/m2 in 25kg/m2

Response: Thanks for the comment. We changed it.

  • Line 115: Please define “biofluids” when it is cited the first time in the main text. Please change “Hossein et al” in “Hosseini et al.”

Response: Thanks for the comment, we added the definition of biofluids in the main text “However, little attention has been given to biofluids (fluids that can be excreted, secreted or develop as a result of a physiological or pathological processes) expression of ncRNAs in the specific setting of pregnancy.”
Also we changed for “Hosseini et al.”

  • Line 133: reference 41: please specifiy

Response: We specified in the text “According to the recent French guidelines, hypertension during pregnancy is de-fined by systolic blood pressure (SBP) > 140 mm Hg and/or diastolic blood pressure (DBP) > 90 mm Hg. When diagnosed before 20 WG, hypertension is considered chronic hy-pertension affecting less than 1% of patients (41).”

  • Line 139: reference 44 and 45 not appear related to the sentence

Response: Thanks for the remark, we deleted references 44 and 45.

  • Line 159: Please specify if the likelihood ratio is 50 or 24

Response: Thanks for the remark, in fact the likelihood ratio is 50.24.

  • Line 176: Mavreli  et  al  change in “Mavreli  et  al.”

Response: Thanks, we changed for “Mavreli et al.”

  • Line 181: Please format properly the table and order the cases by date. Please add data on Nagy et al. (57) or remove it. Add an explanation after the table please. Please change “Not enough materiel” in english.

Response: As suggested, we simplified the table by deleting Nagy et al.

  • Line 209: what (r=- 0.328) stands for?

Response: It stands for “rho=- 0.328” and we modified it in the text.

  • Line 222. Add the point to et al.

Response: We did it for every “et al”

  • Line 394-424: Please add a more specific sentences on the state-of-art of what is discovered now on the topic.

 Response: Thanks for the comment, we clarified it in the text “Among the various pathology associated with pregnancy, only few data are available on miscarriage despite its high incidence, especially for early miscarriage. Moreover, it is not possible to state on their role in clinical routine. Concerning hypertension and preeclampsia, several reports including meta-analysis support the contribution of non-coding RNAs in the diagnosis and the prognosis of PE justifying further studies to evaluate the contribution of miRNAs, circRNAs and lncRNAs to manage patients with risk of hypertension. IUGR represents a crucial issue with major risk for the newborn, the family but also for the society requiring high healthcare resource consumption. Therefore, it should be necessary to evaluate as soon as possible, the inclusion of non-coding RNAs (mainly miRNAs) in biofluids to detect this severe pathology. Due to the increase of obesity in developed countries, GDM represents another major issue. Despite several national and international guidelines to diagnose GDM, the detection of patient is often late thus associated with subsequent risk of increased morbidity for the pregnant woman, the fetus and even the future child. In this setting, it has been demonstrated that miRNAs expression allows to diagnose the prediabetes phase underlining its potential role in clinical routine. Finally, PTB, often the consequence of the aforementioned pathologies, could be decrease by a better screening thanks to the availability of ncRNAs in biofluids, especially in saliva.”

  • Line 423-424: Please add a more specific conclusion on the health care impact of those (potentials) discovers.

Response: Thanks for the comment, we changed it in the text “Finally, the identification of the population at low risk of complications will contribute to a better use of health care resources consumption. Furthermore, as stated by midwifery organizations, the identification of patient at low risk of complications associated with pregnancy could contribute to select patient with a high chance of normal physiologic labor and birth.”

Comments on the Quality of English Language

   The general understanding of English language is good. Hovewer I recommend made a general spell check to reach the level of English of the Journal. 

Response: we suggest sending our paper for correction of the English.

Round 2

Reviewer 2 Report

Comments and Suggestions for Authors

Your work has been carefully examined, and I'm happy to report that it meets our standards and requirements. We appreciate the effort and attention to detail you've put into your submission.

Thank you for your patience throughout this process. 

Author Response

-